# Comparative Effects of Di-(2-ethylhexyl)phthalate and Di-(2-ethylhexyl)terephthalate Metabolites on Thyroid Receptors: In Vitro and In Silico Studies

**DOI:** 10.3390/metabo11020094

**Published:** 2021-02-10

**Authors:** Nicolas Kambia, Isabelle Séverin, Amaury Farce, Laurence Dahbi, Thierry Dine, Emmanuel Moreau, Valérie Sautou, Marie-Christine Chagnon

**Affiliations:** 1Université de Lille, CHU Lille, ULR 7365 GRITA, F-59000 Lille, France; nicolas.kambia-kpakpaga@univ-lille (N.K.); thierry.dine@univ-lille.fr (T.D.); 2Université Bourgogne Franche-Comté, INSERM U1231, NUTOX, Derttech “Packtox”, 21000 Dijon, France; isabelle.severin@agrosupdijon.fr (I.S.); laurence.dahbi@u-bourgogne.fr (L.D.); marie-christine.chagnon@u-bourgogne.fr (M.-C.C.); 3Université de Lille, CHU Lille, INSERM U1286, INFINITE, F-59000 Lille, France; amaury.farce@univ-lille.fr; 4Université Clermont Auvergne, INSERM U1240, IMOST, 63000 Clermont-Ferrand, France; emmanuel.moreau@uca.fr; 5Université Clermont Auvergne, CHU Clermont Ferrand, CNRS, SIGMA Clermont, 63000 Clermont-Ferrand, France

**Keywords:** DEHT, in silico, T-screen assay, hormonal activities, thyroid receptors

## Abstract

Plasticizers added to polyvinylchloride (PVC) used in medical devices can be released into patients’ biological fluids. Di-(2-ethylhexyl)phthalate (DEHP), a well-known reprotoxic and endocrine disruptor, must be replaced by alternative compounds. Di-(2-ethylhexyl) terephthalate (DEHT) is an interesting candidate due to its lower migration from PVC and its lack of reprotoxicity. However, there is still a lack of data to support the safety of its human metabolites with regard to their hormonal properties in the thyroid system. The effects of DEHT metabolites on thyroid/hormone receptors (TRs) were compared in vitro and in silico to those of DEHP. The oxidized metabolites of DEHT had no effect on T3 receptors whereas 5-hydroxy-mono-(ethylhexyl)phthalate (5-OH-MEHP) appeared to be primarily an agonist for TRs above 0.2 µg/mL with a synergistic effect on T3. Monoesters (MEHP and mono-(2-ethylhexyl)terephthalate, MEHT) were also active on T3 receptors. In vitro, MEHP was a partial agonist between 10 and 20 µg/mL. MEHT was an antagonist at non-cytotoxic concentrations (2–5 µg/mL) in a concentration-dependent manner. The results obtained with docking were consistent with those of the T-screen and provide additional information on the preferential affinity of monoesters and 5-OH-MEHP for TRs. This study highlights a lack of interactions between oxidized metabolites and TRs, confirming the interest of DEHT.

## 1. Introduction

Polyvinyl chloride (PVC) is a material widely used in medical devices (MDs), including infusion sets or lines, feeding tubes and tubing, umbilical catheters, oxygen masks, endotracheal tubes, blood transfusers and bags or extracorporeal circuits. Plasticizers are added to the polymer to improve the flexibility and softness of the PVC. However, since they are not covalently bonded to the PVC matrix, they can easily migrate from the MD and come into contact with patients during medical procedures [1,2]. Neonates in intensive care units are known to be exposed to one of these plasticizers, di-(2-ethylhexyl) phthalate (DEHP) present in many MDs [3,4,5]. This phthalate is now classified as a CMR1B (Carcinogenic Mutagenic or Reprotoxic) substance under the Classification Labeling and Packaging (CLP) Regulation [6] due to its effects on reproduction and fertility. The use of DEHP in PVC MDs has been called into question by the European authorities and has been restricted for several years. It must now not exceed 0.1% by mass of the plasticized material, as defined by European Regulation n°2017/745 on MDs [7]. Other plasticizers, such as di-(2-ethylhexyl)terephthalate (DEHT), have been proposed to replace DEHP to soften the PVC in MDs [1]. This additive is particularly interesting because less of it is released from the PVC MDs than DEHP [2], and therefore there is less exposure.

DEHT is less active than DEHP in inducing peroxisome proliferation in rats, which can be explained by the small amount of monoester produced during DEHT metabolism [8]. DEHT is principally hydrolyzed to both terephthalic acid (TPA) and 2-ethylhexanol (EH), with both metabolites being rapidly removed in vivo. This extensive hydrolysis of DEHT to TPA and EH allows only a small fraction to be converted into the monoester and then ultimately to the corresponding oxidized metabolites [9]. MEHT exhibits a lower cytotoxicity than MEHP and its cytotoxicity occurs (0.05 mg/mL) at a much higher concentration than those measured in body fluids [10]. In animal studies, DEHT has shown no reprotoxic effects, a low developmental toxicity and no genotoxicity [9,11].

However, toxicity data are not complete as there is a lack of information regarding the hormonal activities of DEHT and/or its metabolites resulting from in vivo hydrolysis and oxidation. It is, however, very important to assess these activities on hormones since they can occur at very low doses and can have a significant impact on the development of children when they are exposed during critical periods of their development. In a previous study, we performed an in vitro investigation on the potential endocrine-disrupting effects of DEHT and its metabolites on estrogen and androgen receptors and on steroid synthesis. This study demonstrated that DEHT and its metabolites exhibit much weaker effects on hormonal activities than DEHP. However, special attention must be paid to the 5-hydroxy metabolite of mono-(ethylhexyl)terephthalate (5-OH-MEHT) due to co-stimulation of estrogen alpha and human androgen receptors and an increase in estrogen synthesis [12].

To date, data are lacking the effects of DEHT and its ultimate metabolites on thyroid hormonal activity. Phthalates such as DEHP may affect target points in the hypothalamic–pituitary–thyroid axis, such as iodine uptake in the thyroid gland, thyroid hormone synthesis, binding to thyroid hormone receptor or protein transport proteins in blood, biotransformation and excretion and hypothalamic–pituitary control of thyroid hormone production. They may have multiple and possibly overlapping target points, sometimes acting as agonist or antagonist [13]. Ghisari et al. showed that DEHP and other phthalates affected the thyroid hormone-dependent GH3 cell growth using a rat pituitary tumor cell line. A concentration-dependent GH3 cell proliferation was observed with DEHP. Co-treatment of GH3 cells with DEHP and the T3-EC50 inhibited the T3-induced cell proliferation compared to T3-EC50 control [14]. Some studies have demonstrated a correlation between the exposure to DEHP metabolites and thyroid function. The meta-analysis conducted by Huang et al. showed that DEHP exposure can decrease the T4 and increase thyroid-stimulating hormone (TSH). The results suggested that DEHP can affect function in children, adults and pregnant women. Huang and al. concluded that early life phthalate exposure was associated with decreased thyroid hormone levels in young children [15]. In another meta-analysis Kim et al. highlighted that urinary MEHP and 5-OH-MEHP concentration were negatively correlated with total T4, and urinary mono-(2-ethyl-5-oxohexyl)phthalate (5-oxo-MEHP) concentration was positively correlated with thyrotropin [16]. Villanger et al. found that DEHP exposure is associated with thyroid function in mid-pregnancy among Norwegian women. High loadings of DEHP metabolites were associated with a decrease of T3 [14].

These effects may have an impact on neurodevelopment in children. Indeed, the development of the nervous system is extremely dependent on thyroid hormones during the in utero period and the first two years of life, with critical windows of vulnerability [17,18,19,20]. Newborns and premature neonates hospitalized in intensive care units are particularly vulnerable. It is therefore very important to assess whether plasticizers from MDs, and the corresponding metabolites found in the body, affect the thyroid and the active concentrations.

The objective of this study was to use in silico and in vitro methodologies to assess the effects of DEHT metabolites on thyroid hormonal activities and to compare them with those of DEHP, as both plasticizers are highly metabolized.

## 2. Results

### 2.1. Access to the Secondary Metabolites 5-OH-MEHT, 5-oxo-MEHT and 5-cx-MEHT

Metabolites 5-OH-MEHT, 5-oxo-MEHT and 5-cx-MEHT, mono-(2-ethyl-5-carboxypentyl)terephthalate, were synthesized from 2-ethylhex-5-en-1-ol (**1**) and 4-((benzyloxy)carbonyl)benzoic acid (**2**), previously synthesized and characterized (^1^H, ^13^C NMR and HRMS) by Nüti et al. and our lab INSERM U1240 [10,21], respectively (Scheme 1). Derivatives (**4**) and (**7**) were obtained by Wacker oxidation of the vinylic group at ω-position of compound (**3**) in a mixture of PdCl_2_ and parabenzoquinone or a hydroboration reaction, respectively. Compound (**4**) was then converted into 5-oxo-MEHT (**5**) by hydrogenation, with the ketone group then being reduced with NaBH_4_ to form 5-OH-MEHT (**6**). Finally, compound (**7**) was successively oxidized (Jones reagent) and reduced (Method A; or inverse for Method B) to give 5-cx-MEHT (**9**). The purity of all synthesized metabolites was over 95% (HPLC/MS).

Metabolites 5-OH-MEHT, 5-oxo-MEHT and 5-cx-MEHT, mono-(2-ethyl-5-carboxypentyl)terephthalate, were synthesized from 2-ethylhex-5-en-1-ol (**1**) and 4-((benzyloxy)carbonyl)benzoic acid (**2**), previously synthesized and characterized (^1^H, ^13^C NMR and HRMS) by Nüti et al. and our lab INSERM U1240 [10,21], respectively (Scheme 1). Derivatives (**4**) and (**7**) were obtained by Wacker oxidation of the vinylic group at ω-position of compound (**3**) in a mixture of PdCl_2_ and parabenzoquinone or a hydroboration reaction, respectively. Compound (**4**) was then converted into 5-oxo-MEHT (**5**) by hydrogenation, with the ketone group then being reduced with NaBH_4_ to form 5-OH-MEHT (**6**). Finally, compound (**7**) was successively oxidized (Jones reagent) and reduced (Method A; or inverse for Method B) to give 5-cx-MEHT (**9**). The purity of all synthesized metabolites was over 95% (HPLC/MS).

### 2.2. Impact on Thyroid-Dependent Cell Growth

The T-screen assay was used as a fast and functional assay to assess interference with T3 receptors (agonistic or antagonistic potency of xenobiotics at cellular level (Figure 1)).

A concentration-dependent antagonist effect starting at 2 µg/mL was observed, becoming significant at 5 µg/mL with MEHT. Above 5 µg/mL, MEHT was cytotoxic for the cell line. MEHT metabolites had no effect.

In contrast, MEHP was a partial agonist (resulted in a small but significant increase of cell growth (<15%) between 10 µg/mL and 20 µg/mL. Derived oxidative metabolites, such as OH metabolite, were 2-fold more active, with a significant effect at lower concentrations (0.2 µg/mL, 2 µg/mL and 5 µg/mL) and with a concentration dependency. A synergistic response was also observed when cells were co-treated with T3 up to a concentration of 10 µg/mL. 5-oxo-MEHP significantly inhibited cell growth at 10 µg/mL in the presence of T3 and at non-cytotoxic concentrations. Higher concentrations of 5-oxo-MEHP were very cytotoxic for the cells. The CX metabolite had no effect.

### 2.3. Docking

T3 was docked into the two subtypes to verify the docking protocol. A single conformation was achieved, which was almost superimposable with the co-crystallized conformation. It was tightly bound by a strong ionic interaction between the acid and arginine 228 and 262 at the base of the pocket. At the other extremity, the phenol formed a hydrogen bond with His381. The only difference was a slight twist of the acid chain to better interact with the arginines (Figure 2).

DEHP and DEHT barely fit into the pocket. It was clearly a misfit due to the sheer volume of the pocket, which was able to accommodate the bulk of the diesters. However, it was not possible to find a single pose, as the aromatics were only loosely positioned in the same area of the pocket, but absolutely not in a preferred conformation and fully lacking interactions with the receptor. This was more marked for DEHT than for DEHP, although common to the two molecules.

The monoesters fared well. MEHT showed a single conformation, with an excellent conservation of the position of the aromatic ring close to arginines 228, 262 and 266, which interacted with its free acid via strong salt bridges. The remaining ester side chain occupied the other side of the pocket and, with only a small amount of fluctuation, was positioned above His 381. Compared to T3, the aromatic group of MEHT was at the opposite end of the pocket. MEHP, in contrast, fitted into the binding site with its aromatic group very close to the position of the distal phenyl of the hormone. The free acid was also able to bind to His 381, although the orientation of the interaction was not perfect in the crystallographic conformation of the histidine side chain. The ester occupied the part of the pocket close to the arginines (Figure 3).

The hydroxylation of the monoester metabolite in 5-OH-MEHT did not modify its placement in the pocket. However, the hydroxyl at the end of the ester chain only formed inconsistent hydrogen bonds with the skeleton of Gly290, which may be deleterious due to the rather hydrophobic nature of the pocket around this residue. 5-OH-MEHP behaved in much the same way, keeping the same position as its parent but with the acid binding to His381 and the hydroxyl group at the other end of the molecule forming hydrogen bonds with the skeleton of Met259 and, occasionally, with that of Ala283 (Figure 4).

The docking results were therefore fairly comparable so the discussion of the docking will focus on TRα, which benefits from a better resolution. The conclusions drawn are exactly the same for TRβ1.

## 3. Discussion

### 3.1. Impact on Thyroid Hormones

Thyroid hormones have a wide range of biological effects in vertebrates, during both fetal and prenatal development and with regard to the development of sex organs and the central nervous system in mammals [22,23]. The T-screen is based on thyroid hormone-dependent cell growth of a rat pituitary tumor cell line and is used as a model to study basic thyroid hormone-dependent cell physiology, and to study the interference of compounds with thyroid hormones at a cellular level [24,25].

For the DEHT metabolites, only MEHT was an antagonist for cell line proliferation at very low concentrations. In contrast, DEHP metabolites were agonists, particularly the hydroxylated metabolite (5-OH-MEHP), demonstrating a synergism when cells were co-treated with T3 up to a concentration of 10 µg/mL. 5-oxo-MEHP only inhibits cell growth at 10 µg/mL and was very cytotoxic for the cells above this concentration. Our data agree with data published by Ghisari that demonstrated a low dependent potency activation of DEHP between 10^−6^ M and 10^−5^ M. However, Ghisari et al. did not examine DEHP metabolites [14]. In a TR reporter gene assay using a recombinant *Xenopus laevis* cell line, benzyl butyl phthalate (BBP), dibutyl phthalate (DBP) and DEHP were reported to exhibit a T3-antagonistic activity, and to inhibit the expression of the endogenous TRβ gene [26]. DEHP was also shown to interfere with the binding of T3 to TRβ [27], suggesting that the compound may bind to the receptor. Rodent studies have also confirmed the effects of DEHP on the thyroid [28,29], where significant influences on thyroid hormones and metabolism were observed. Specifically, proliferative changes were noted in the thyroid in vivo, raising the concern of a potential thyroid carcinogenicity of DEHP. Recently, Kim et al. investigated whether DEHP could induce proliferative changes and DNA damage in 8505C thyroid carcinoma cell lines, both in vitro and in the thyroid tissue of rats treated orally with DEHP for 90 days from juvenile to full maturation in vivo [16]. They showed that DEHP can stimulate thyroid cell proliferation and DNA damage through the activation of the TSHR pathway, as TSHR plays a key role in the proliferation and differentiation of thyroid cells [30]. All in vitro and in vivo data suggested that DEHP is able to influence thyroid tissues at low doses.

Although studies on the in vitro effects of plasticizer metabolites on the thyroid hormone (TH) system are scarce, it is well known that diesters are highly metabolized in vivo. Reduced serum thyroid hormone levels have been well documented in human populations, with higher urine phthalate metabolites in various regions around the globe [31,32,33,34].

In this study, MEHP and 5-OH-MEHP appear partial agonists on the GH3 proliferation, 5-OH-MEHP being more active. These data are in accordance with Ghisary et al.’s [14] work showing that DEHP can induce the proliferation of the GH3 cells. However, they did not check if the DEHP hydrolysis can occur in the experimental conditions. We observed that 5-OH-MEHP, but not MEHP, also induced a decrease in the presence of T3 at the highest non-cytotoxic concentration, whereas below this concentration we saw a synergism at 5 µg/mL. In this assay, cell proliferation was measured as a consequence of T3 activation, but the cell proliferation is a complex biological phenomenon, not just TR-dependent. For example, GH3 cells also express peroxisome proliferator-activated receptors (PPARs), which exhibit anti-proliferative activity upon ligand binding [35]. In this study, we can assume that the observed response is linked to T3, as induction by T3 was modulated in the presence of 5-OH-MEHP when the cells were co-exposed. Interestingly, human studies have found an inverse association between MEHP metabolites in the urine and concentrations of free T4 and total T3 levels in adult men [33], which suggests that DEHP can disrupt the homeostasis of the thyroid–pituitary axis.

### 3.2. Docking

DEHP and DEHT were unable to bind to TRα, only occupying the pocket due to a volume barely large enough to accommodate them. It can be concluded that they are not able to directly interact with TRs. On the contrary, the monoesters MEHP and MEHT displayed a rather strong binding mode in the pocket, with a single coherent conformation found for each. It is noteworthy that MEHP, with its aromatic group in close proximity to His381, closely imitates the binding mode of the natural T3 agonist despite lacking any strong interaction with the arginines. On the other hand, MEHT, presenting its acid at the entry of the pocket in front of the arginines, has a different binding mode in which the aromatic is at the opposite end to His 381 and has no interactions with this part of the binding site.

The presence of a hydroxyl group on the monoester metabolite did not alter its placement in the pocket, with 5-OH-MEHP once again being closer to the natural agonist and 5-OH-MEHT positioning its aromatic group the other way around, far from His381. This conformation is also somewhat destabilized by the poor hydrophobic fit between the added hydroxyl group and the rather hydrophobic area around His381.

Further oxidation of the hydroxyl to a carbonyl group led to the same positioning of the derivatives. In the case of 5-oxo-MEHP, the same position close to His381 was maintained, and the carbonyl formed some hydrogen bonds, although it did not exhibit a clear preference, binding to Arg228, the Ser277 skeleton or side chain, or to nothing, with roughly the same propensity. It may therefore be a less than perfect fit for the TR binding site and clearly inferior to the hydroxylated metabolite. The carbonyl group of 5-oxo-MEHT formed no interactions and lay in the middle of the pocket with a poor fit to its surroundings.

Interestingly, of the 30 solutions determined for 5-cx-MEHP, no correctly superimposable conformation was identified. Although all were placed in the same area, there was a good deal of fuzziness in their position, most probably due to the different relative positions of the acids in T3 and in 5-cx-MEHP, hindering a perfect fit to the binding site. The addition of a second acid at the end of the remaining ester on 5-cx-MEHT gave the same position as the other compounds derived from DEHT. The acid of the ester side chain lay squarely in the middle of the pocket and was unable to form any interactions. We can therefore assume that it is less capable of binding than its congeners.

Table 1 summarizes the theoretical and experimental affinities toward TRα1 obtained in silico and in vitro.

### 3.3. In Vitro Data versus Biomonitoring Values

In neonatal intensive care units (NICUs), the use of many plasticized PVC medical devices overexposes neonates to phthalates [3,4,5]. Biomonitoring studies have shown high concentrations of oxidized metabolites of DEHP in the urine of neonates. Urinary levels of 5-OH-MEHP may exceed 0.2 µg/mL, a concentration at which we have demonstrated a TRβ agonist effect that was confirmed in vitro as being partial and has a synergistic effect with T3. The maximum values of 5-OH-MEHP measured in the urine of newborns hospitalized in NICUs range from 0.43 µg/mL to 13.1 µg/mL, i.e., 2 to 65 times higher than the concentration activating thyroid receptors [36,37,38,39,40]. Certain medical procedures, such as extracorporeal circulation (extracorporeal membrane oxygenation, cardiopulmonary bypass), respiratory assistance, and intravenous nutrition are recognized as situations with a high risk of exposure, which can explain the high values in multi-exposed patients. In the previously cited studies, the median values of urinary 5-OH-MEHT concentrations are generally less than 0.2 µg/mL. Calafat et al. (2004) [38] and Green et al. (2005) [39] showed higher urinary levels of 2.22 µg/mL and 0.26 µg/mL, respectively. However, these studies were carried out in 2004 and 2005, and it can be assumed that exposure to DEHP via MDs has decreased in the 15 years following the recommendations of the current Scientific Committee on Emerging and Newly-Identified Health Risks, SCENIHR, and the European regulations. However, a recent study carried out specifically in newborns in cardiac surgery showed that extracoporeal circulation medical devices remain highly exposed to DEHP. Indeed, Gaynor et al. (2019) [41] demonstrated that the level of 5-OH-MEHP passes from 0.01 µg/mL to 0.229 µg/mL after cardiopulmonary bypass. Our study showed that MEHP had partial agonist effects in the T-screen bioassay, suggesting an action on TRs at concentrations of 10 µg/mL to 20 µg/mL. The data in the literature show that the concentrations of MEHP in biological media are significantly lower than these values, including the study by Eckert et al., who directly measured MEHP in blood in contact with extracorporeal circulation lines during heart surgery in newborns. The maximum concentration of MEHP found in the blood of these patients was 0.56 µg/mL [42].

With regard to DEHT metabolites, only MEHT showed an effect on cellular growth with an antagonistic effect on T3 at non-cytotoxic concentrations of 2 µg/mL to 5 µg/mL using the T-screen assay. There is little biomonitoring data for DEHT in the literature, even less measuring the exposure of hospitalized newborns. Lessmann et al. studied the exposure of non-hospitalized children over 4 years of age to DEHT [43]. However, only the oxidized metabolites of MEHT were measured in the urine. Pinguet et al. presented results of the exposure of patients hospitalized in the NICU to certain plasticizers, including DEHT [40]. The median of the urinary concentrations of MEHT in this cohort of patients was lower than the limit of quantification (0.018 ng/mL) and the maximum concentration measured was 9.90 ng/mL, 200 times less than the concentration showing an antagonistic effect on T3 identified using the T-screen assay in this study [40]. In the Armed Neo clinical trial, MEHT levels found in the urine of premature babies hospitalized in the NICU were also much lower than this value, with a maximum of 1.32 ng/mL (unpublished data). Therefore, it would appear unlikely that MEHT levels of 2 µg/mL would be reached in the biological media of hospitalized newborns. DEHT is a plasticizer that has a low migration from PVC medical devices [2,44]. In addition, MEHT, a metabolite resulting from the enzymatic hydrolysis of DEHT, is very rapidly transformed into oxidized derivatives, which have no in vitro effect on T3-dependent cell proliferation. The urinary excretion factor is 0.02% and 6% for MEHT and MEHP, respectively [45].

## 4. Materials and Methods

### 4.1. Metabolites of DEHP and DEHT

Primary and secondary metabolites of DEHP and DEHT were synthesized and characterized by the IMOST team (UMR 1240, INSERM) in Clermont-Ferrand, France. The compounds tested are shown Table 2. The purity of all our synthesized metabolites and their corresponding intermediates exceeded 95% ((High performance liquid chromatography with mass spectrometry (HPLC/MS) and Nuclear magnetic resonnance (NMR)). MEHT was synthesized according to the method described by Eljezi et al. [10]. All synthesis processes are described in the supplementary data (Appendix A).

### 4.2. Preparation of Samples

All compounds were dissolved in 100% ethanol and tested in a concentration range of 0.2 µg/mL to 20 µg/mL. The maximum concentration of ethanol in the culture medium was 0.1% (vol/vol) in order to avoid any cytotoxic effect of the vehicle.

### 4.3. T-Screen Assay

#### 4.3.1. Cell Culture and Treatment

The assay was based on thyroid hormone-dependent cell growth of the rat pituitary tumor cell line GH3 (ATCC, CCL-82.1). The GH3 cells were cultured at 37 °C in a humidified atmosphere with 5% CO_2_ in phenol-red-free DMEM/F12 (Gibco-Invitrogen, Fisher Scientific, Illkrich, France) supplemented with 10% (vol/vol) fetal bovine serum (PAN, Biotech, Dutscher, Brumath, France). Passaging was carried out in 75 cm^2^ tissue culture flasks (Falcon) every four days by releasing the cells from the substrate using 0.25% (*w*/*v*) trypsin/1 mM EDTA solution (PAN, Biotech, Dutscher, Brumath, France). The T-screen was performed as previously described [46]. GH3 cells at 80% confluence were incubated for 48 h in serum-free PCM medium (which was changed once after 24 h), as originally described by Sirbasku et al. [47]. PCM consisted of phenol-red-free DMEM/F12 with 15 mM HEPES (Gibco-Invitrogen, Fisher Scientific, Illkrich, France), 10 μg/mL bovine insulin (Sigma- Aldrich, St Quentin Fallavier, France, T6634), 10 μM ethanolamine (Sigma- Aldrich, St Quentin Fallavier, France, E0135), 10 ng/mL sodium selenite (Sigma- Aldrich, St Quentin Fallavier, France, S5261), 10 μg/mL apo-transferrin (Sigma- Aldrich, St Quentin Fallavier, France, T1147), and 500 μg/mL bovine serum albumin (Sigma- Aldrich, St Quentin Fallavier, France, A7906). The cells were then harvested in PCM medium using a cell scraper and plated at a density of 2500 cells/well (100 µl) on a 96-well plate (Falcon, Dutscher, Brumath, France). Following an attachment period of 2 h to 3 h, the cells were exposed in triplicate and for 96 h to various concentrations of the chemicals to be tested (100 µL, 2× dosing exposure concentration in PCM medium), either alone or in combination with a median effective concentration of 0.25 nM T3 (CAS: 6893-02-3, Sigma-Aldrich, St Quentin Fallavier, France, T2877). Negative control wells contained cells and test medium with the same amount of ethanol (0.1%) as the exposed cells. Positive controls included were T3 (agonist control) and amiodarone (antagonist control, CAS: 19774-82-4, Sigma-Aldrich, St Quentin Fallavier, France, A8423). They were used as a dose response control in the assay (from 0.001 to 10 nM for T3 and from 0.25 to 500 nM for amiodarone) to validate it for detection of agonistic and antagonistic activities (Appendix B).

#### 4.3.2. Cytotoxicity/Viability/Proliferation

Following a 4 h incubation period with 10 μL/well of 0.1 mg/mL resazurin (Sigma-Aldrich, St Quentin Fallavier, France) in PBS, cell proliferation was measured as relative fluorescence units (RFUs) resulting from the reduction of non-fluorescent resazurin to the fluorescent product resorufin. Fluorescence, a measure of the amount of viable cells present, was recorded at λ_ex_ = 530 nm and λ_em_ = 590 nm on a microplate reader (Chameleon from Hidex, Finland). The assay can detect antagonist effects (inhibition of cell proliferation) measured by a fluorescence decrease. However, it is difficult to distinguish an inhibition of cell proliferation from a cytotoxicity as both effects are expressed by a decrease in fluorescence. So concentrations of compounds that inhibited basal metabolic activity of GH3 in the agonist assay were excluded from statistical analysis. A chemical was considered cytotoxic if the fluorescence was less than the fluorescence of the vehicle control minus 3-fold the standard deviation in the agonist assay.

#### 4.3.3. Data Analysis

Cell proliferation was expressed as a function of the maximum response observed at 10 nM T3 (in agonist mode) or 0.25 nM T3 (in antagonist mode), which was set at 100% induction. The response for the solvent control was set at 0%.

#### 4.3.4. Statistical Analysis

Obtained data were statistically analyzed using the PC program GraphPad Prism 6.00 (GraphPad Software Incorporated, San Diego, CA, USA). Descriptive statistical characteristics (arithmetic mean, minimum, maximum, standard deviation and coefficient of variation) were evaluated. One-way analysis of variance (ANOVA) and the Dunnett’s multiple comparison test were used for statistical evaluations. The level of significance was set at *** *p* < 0.001; ** *p* < 0.01 and * *p* < 0.05.

### 4.4. Docking

DEHP, DEHT and their respective metabolites were docked into TRα1, the only TRα subtype to bind T3, and TRβ1, the only β subtype crystallized. The coordinates of the receptor subtypes were taken from the RCSB ProteinDatabank under the entry 4lnw [48] crystallized with T3, and 1n46 [49], bound to an agonist, respectively. The ligands were extracted manually and both molecules were assigned using the Gasteiger–Hückel method. The ligands were subjected to an energy minimization using the maximin2 protocol of the Sybyl 6.9.1 molecular modeling package. The co-crystallized T3 was cross-docked with very good precision, indicating that the docking protocol was sound. It consisted of a 30-solution GOLD run in a binding site defined as a sphere of 10 Å around the co-crystallized ligand with 100,000 operations and the ChemPLP scoring function. The 30 poses were manually inspected to define the most representative conformations, chosen as the best scored solution from the largest cluster of poses. In a few cases, a high score, but not the best, was selected as it was more representative of all the poses. The two receptor subtypes only differed in 15 residues, all of which were remote from the binding site. The superimposition of the structures was also fairly good, with a Root Mean Square over the heavy atoms of 2.42 Å, which is only very slightly higher than the worst resolution (2.2 Å). The docking results were therefore fairly comparable, so the discussion of the docking focused on TRα, which benefited from a better resolution.

## 5. Conclusions

We can conclude that under these experimental conditions, and regard to the use of alternative methods (in vitro, in silico), that DEHT-oxidized metabolites have negligible effect on T3 hormonal activities when compared to DEHP metabolites. Taking all these data into account, along with human biological enzymatic data, there appears to be no safety concern with DEHT compared to DEHP tested in the same conditions and when based on a mode of action.

## Data Availability

Data is contained within the article.

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
