# Peer review of "Comparative Effects of Di-(2-ethylhexyl)phthalate and Di-(2-ethylhexyl)terephthalate Metabolites on Thyroid Receptors: In Vitro and In Silico Studies"

_metabolites, 2021, doi:10.3390/metabo11020094_

Round 1

Reviewer 1 Report

The authors conducted in silico and in vitro studies on the effects of metabolites of DEHT and DEHP on thyroid hormone receptor. The rationale for these studies is that DEHT is being considered as an alternative to DEHP in polyvinyl chloride used in medical devices, because DEHP has been reported to be an endocrine disruptor. This field of study is very significant in the current times. The authors demonstrate that DEHP metabolites are agonistic to thyroid hormone receptor beta, wheres the oxidized metabolites of DEHT do not have significant interaction with this receptor. These results will have an impact as they support that DEHT could hence be considered as a safer alternative to DEHP in medical devices. There are a few minor changes that I suggest: 

  • A part of the results are presented before the Materials and Methods section. All the results should be presented together in the Results section. 
  • On page 12, there is a paragraph mentioning the instructions for how to reports Materials and Methods. This is presumably part of author instructions which was copied to the manuscript by mistake, so should be removed. 
  • The Discussion should be presented after the Materials and Methods section. 

Author Response

1-A part of the results are presented before the Materials and Methods section. All the results should be presented together in the Results section. 

We have followed the author instructions recommanding to present the results before materials and methods. However we have transfered the chapter 4.2 initially present in the section Materials et Methods in the section « Results » as chapter 2.1.

See page 3 lines 109-121

2-On page 12, there is a paragraph mentioning the instructions for how to reports Materials and Methods. This is presumably part of author instructions which was copied to the manuscript by mistake, so should be removed. 

Sorry for this mistake. We have removed this part of author instructions

3-The Discussion should be presented after the Materials and Methods section

We have respected the author instructions concerning the order of the chapters. It’s the raison why the discussion appears before materials and methods

Reviewer 2 Report

The authors compared effects of DEHP and DENT including their metabolites MEHP, MENT, 5-OH MEHP, and 5-OH MENT on the thyroid receptor TRbeta. The authors simulated the docking of the substrates compared to T3 and used the established T screen assay in absence and presence of T3 to assess functional activity. Motivation of the study, methodology, and results are clearly described and discussed. At few places, more discussion would be needed.

From the text, docking to the TRa1 is described but the binding to TRb1 is not clear.

Phthalates interfere with thyroid function on different levels and the testing assesses only a part of the potential effects of MENP. Action of phthalates on thyroid metabolism should be mentioned in the Introduction and the limitation that the presented data addressed only part of the possible effects in the Conclusions.

Minor

l.359-365 can be removed; they are instructions from the template.

Author Response

1-From the text, docking to the TRa1 is described but the binding to TRb1 is not clear.

There were no significant differences between the docking in the two subtypes and the binding sites are identical. If there are differences, they will be at an earlier stage, before the ligand has arrived into the binding pocket, and are therefore not discernible with a purely docking study. We only described the results obtained for TRa1 as duplicating them for TRb1 was felt unnecessary. However, the manuscript has been modified to make this clearer.

See page 6 lines 171-172

2-Phthalates interfere with thyroid function on different levels and the testing assesses only a part of the potential effects of MENP.

In the introduction we have given more details concerning the interference of phthalates, particularly DEHP on thyroid function.

See page 2-3 lines 81-98

3-Minor

l.359-365 can be removed; they are instructions from the template

Sorry for this mistake. We have removed this part of author instructions

Reviewer 3 Report

The authors reported an alternative method to animal studies to assess the activities of DEHT and DEHP on T3 hormonal by integrating the in vitro and silico methods. I believe that the authors did a good job and made a great contribution to this field. I just have one suggestion. Please separate the introduction into at least three paragraphs because it is difficult to read by the reader. 

Author Response

The authors reported an alternative method to animal studies to assess the activities of DEHT and DEHP on T3 hormonal by integrating the in vitro and silico methods. I believe that the authors did a good job and made a great contribution to this field. I just have one suggestion. Please separate the introduction into at least three paragraphs because it is difficult to read by the reader.

We have separated the introduction on different paragraphs

See page 2 and 3